



# Technical note: Pitfalls in using log-transformed flows within the KGE criterion

Léonard Santos, Guillaume Thirel, and Charles Perrin

Irstea, HYCAR Research Unit, 1 rue Pierre-Gilles de Gennes, 92160 Antony, France

*Correspondence to:* Léonard Santos (leonard.santos@irstea.fr)

**Abstract.** Log-transformed discharge is often used to calculate performance criteria to better focus on low flows. This prior transformation limits the heteroscedasticity of model residuals and was largely applied in criteria based on squared residuals, like Nash-Sutcliffe efficiency (NSE). In the recent years, NSE has been shown to have mathematical limitations and Kling-Gupta efficiency (KGE) was proposed as an alternative to provide more balance between the expected qualities of a model
(namely representing the water balance, flow variability and correlation). As in the case of NSE, several authors used the KGE criterion (or its improved version KGE') with a prior logarithmic transformation on flows. However, we show that the use of this transformation is not adapted to the case of the KGE (or KGE') criterion and may lead to several numerical issues, potentially resulting in a biased evaluation of model performance. We present the theoretical underpinning aspects of these issues and concrete modelling examples, showing that KGE' computed on log-transformed flows should be avoided. Alternatives are
discussed.

## 1 Introduction

In the context of rainfall-runoff modelling, evaluating the quality of the models' outputs is essential. Deterministic simulations are commonly evaluated using efficiency criteria such as Nash-Sutcliffe efficiency (NSE, Nash and Sutcliffe, 1970). The choice of the criteria obviously depends on the modeller's objective. For example, one may wish to focus on the overall water balance
evaluation, or more specifically on the simulation of different flow ranges, typically high, intermediate or low flows. For these different objectives, given that the model residuals are generally not homoscedastic and often depend on the flow magnitude, one common option to focus more closely on specific flow ranges is to apply various prior transformations on the simulated and observed discharge time series to distort the range of errors, which consequently changes the relative weight of different flow ranges in the criterion. This is commonly done within the NSE criterion, which has been one of the most popular criteria
used in hydrological modelling in the past few decades. NSE is the distance to 1 of the ratio between the mean square error of the model and the variance of observed flows. Compared to the basic criterion computed on untransformed flows, a prior squared transformation on flows would put even more weight on high flows, a logarithmic or inverse transformation would put more weight on low flows while a square root transformation would have an intermediate effect (Krause et al., 2005; Oudin et al., 2006; De Vos and Rientjes, 2010; Pushpalatha et al., 2012).





However, the Nash-Sutcliffe criterion was shown to have limitations. Indeed, using a decomposition of NSE based on the correlation, bias and ratio of variances, Gupta et al. (2009) clearly demonstrated that discharge variability is not correctly taken into account for the evaluation. Therefore, Gupta et al. (2009) proposed a new criterion, Kling-Gupta efficiency (KGE), which was then improved into a modified criterion called KGE' (Kling et al., 2012). KGE combines the previous components of NSE

(correlation, bias, ratio of variances or coefficients of variation) in a more balanced way. It corrects the underestimation of variability and provides direct assessment of four aspects of discharge time series, namely shape and timing, water balance and variability.

Given that this criterion tends to be sensitive to large errors, some users chose to apply prior transformations on flows before computing KGE, e.g. to put more weight on low flows, as done with NSE. For example, Pechlivanidis et al. (2014) applied

the logarithmic transformation to use it as a benchmark for fitting a model on low flows. Seeger and Weiler (2014) used it as an objective function. Beck et al. (2016) used the untransformed and log-transformed flows in NSE, $R^2$ and KGE as an evaluation of different global models, and Quesada-Montano et al. (2018) also used it as an evaluation criterion of the HBV model outputs.

In this technical note we show that the use of a logarithmic transformation when computing KGE or KGE', applied in a

similar way as with NSE, introduces numerical flaws and should be avoided. After reviewing the mathematical formulation of KGE', we expose the theoretical aspects explaining these flaws and illustrate them with modelling examples. Then we suggest alternatives to circumvent this issue. The tests will be carried out using KGE' but they are also valid for the initial KGE formulation.

## 2   The KGE' formulation

The KGE' criterion (Kling et al., 2012, denoted $E_{KG}$ in Eq. 1) is written as a sum of the distances to 1 (perfect value) of three components of the modelling error:

$$E_{KG} = 1 - \sqrt{(r-1)^2 + (\beta-1)^2 + (\gamma-1)^2} \qquad (1)$$

in which:

- $r$, the Pearson correlation coefficient, evaluates the error on shape and timing between observed ($Q_o$) and simulated ($Q_s$)

flows:

$$r = \frac{\mathrm{cov}(Q_o, Q_s)}{\sigma_o^2 \sigma_s^2} \qquad (2)$$

- $\beta$, the bias term, evaluates the water balance error:

$$\beta = \frac{\mu_s}{\mu_o} \qquad (3)$$





– $\gamma$, the ratio between the simulated and observed coefficients of variation (CV) evaluates the flows variability error:

$$\gamma = \frac{\mu_\mathrm{o}\sigma_\mathrm{s}}{\sigma_\mathrm{o}\mu_\mathrm{s}} \tag{4}$$

where $\mathrm{cov}$ is the covariance between observation and simulation, $\mu$ is the mean and $\sigma$ is the standard deviation, with subscripts $o$ and $s$ standing for observed and simulated, respectively.

The KGE' values range between $-\infty$ and $1$, as for NSE, and it is positively oriented.

## 3 Issues associated with the use of a prior logarithmic transformation

### 3.1 Instability when the moments of log-transformed flows become close to zero

Because the three terms, $\gamma$, $\beta$ and $r$ are ratios, they can become overly sensitive to the denominator values (here $\mu_\mathrm{o}$, $\mu_\mathrm{s}$, $\sigma_\mathrm{o}$ or $\sigma_\mathrm{s}$) if they become close to zero. In this case, a small absolute variation in the moments' values can negatively impact the

related ratio and thus produce very negative KGE' values. It is generally very unlikely to obtain values of $\sigma_\mathrm{o}$, $\sigma_\mathrm{s}$, $\mu_\mathrm{s}$, $\mu_\mathrm{o}$ so close to zero to produce numerical instability when using untransformed flows. However, when a prior logarithmic transformation is applied, the values of $\mu_\mathrm{log,o}$ or $\mu_\mathrm{log,s}$ (more rarely $\sigma_\mathrm{log,o}$ or $\sigma_\mathrm{log,s}$) computed on transformed values can become equal or very close to zero (because $\log(1) = 0$). The corresponding ratios $r$, $\beta$ or $\gamma$ would therefore become very large, leading to strongly negative KGE' values. Thus a small relative difference can lead to very different conclusions. In this case, the score value does

not adequately represent the qualities of the model simulation.

### 3.2 Dependence on the flow unit chosen

KGE' and NSE criteria are dimensionless. This means that using discharge values expressed in litres per second or in cubic metres per second has no impact on the criteria values. It can be easily demonstrated that $\gamma$, $\beta$ and $r$ remain identical when flow is expressed in any of these two units, since the division by 1000 necessary for the conversation is eliminated in the ratios.

When using a prior logarithmic transformation, the NSE criterion is not affected because the squared differences of flows eliminates the multiplicative conversion coefficients in the mean square error (numerator) or in the variance (denominator). However, the KGE' calculation is altered through the $\beta$ ratio. Using the example of the average observed flow calculation, the conversion from cubic metres per second to litres per second gives the following:

$$\mu_\mathrm{log,o}[\mathrm{l} \cdot \mathrm{s}^{-1}] = \log(1000) + \mu_\mathrm{log,o}[\mathrm{m}^3 \cdot \mathrm{s}^{-1}] \tag{5}$$

Consequently, because the conversion term becomes additive when applying the logarithmic transformation, the $\beta$ ratio value is modified. Similarly, the $\gamma$ ratio is also altered.

Therefore, if the logarithmic transformation is used, the KGE' (and also the KGE) is no longer a dimensionless value. This can lead to interpretation problems.





### 3.3 Dependence on the constant added to avoid the zero-flow issue

When using a logarithmic (or an inverse) transformation, the case of null flows, which may exist in case of intermittent or ephemeral streams, prevents proper calculation. To avoid this, different techniques may be set up in the case of NSE:

- discarding the zero-flow values from the series, i.e. considering them as gaps (see e.g. Nguyen and Dietrich, 2018). The drawback is that parts of the hydrographs become neglected, though they can bring important information on the processes at play.

- using a Box-Cox transformation to reproduce the effects of the logarithmic transformation without the zero-flow issue (Box and Cox, 1964; Hogue et al., 2000; Vázquez et al., 2008).

- adding a small constant to all flow values (Pushpalatha et al., 2012), typically a fraction of average flow. This option is widely used and Pushpalatha et al. (2012) showed that the NSE value has limited sensitivity to this constant with a logarithmic transformation as long as it is small enough compared to flow values. These authors advise a constant equal to one-hundredth of the mean observed flows. But the dependence of KGE' on this constant has not been investigated so far.

## 4 Testing methodology

To illustrate these numerical issues and their potential impacts, several tests were made on a wide range of catchments, using the GR4J rainfall-runoff model (Perrin et al., 2003).

### 4.1 Catchment set and data

A daily data set of 240 catchments across France (Fig. 1), set up by Ficchí et al. (2016), was used. The climate data of the SAFRAN daily reanalysis (Vidal et al., 2010) were used as input data. Precipitation and temperature were spatially aggregated on each catchment since the GR4J model is lumped. Potential evapotranspiration was calculated using a temperature-based formula (Oudin et al., 2005). Full details on this data set are available in Ficchí et al. (2016).

Observed flows were retrieved for each catchment outlet from the *Banque HYDRO* (http://www.hydro.eaufrance.fr/, Leleu et al., 2014). The availability of data covers the 2003-2013 period.

The catchments were selected to have less than 10% of precipitation falling as snow, to avoid requiring a snow model.

### 4.2 Model and calibration

The tests were performed with the daily lumped conceptual GR4J model (Perrin et al., 2003).

The four parameters of the model are calibrated using the local search optimization algorithm used in Coron et al. (2017). The available records are split into a calibration (2003-2008) and a validation (2008-2013) period following a standard split-sample test procedure (Klemeš, 1986). The calibration procedure was run using the KGE' on untransformed flows as an objective



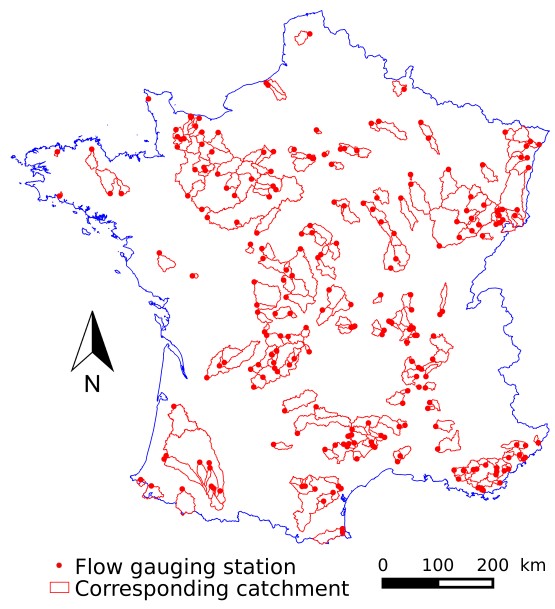

**Figure 1.** Location of the 240 flow gauging stations in France used for the tests and their associated catchments.

function. The performance of the model is then evaluated during the validation period using KGE' on untransformed and log-transformed flows. The performance is also calculated using different transformations that can substitute the logarithmic transformation, namely the square-rooted flows, the inverted flows and the Box-Cox transformed flows. The NSE criterion is also calculated on log-transformed flows to be compared to KGE' using the same transformation. The zero flows were treated

5 following the conclusions of Pushpalatha et al. (2012), i.e. by adding to flows a constant equal to one-hundredth of the mean observed flows. The parameter of the Box-Cox transformation is fixed at the value of 0.25, as Vázquez et al. (2008) argue that it is an usual value in hydrological studies.

## 5 Results

### 5.1 Instability when the moments of log-transformed flows become close to zero

10 Figure 2 analyses the stability of the KGE' values with log-transformed flows obtained in the validation period. The KGE' values were plotted against the mean of the log-transformed observed (a) and simulated (b) flows. When any of these means tends to be close to zero, the KGE' criterion exhibits unusually low values. This plot illustrates the problem identified in section 3.1. These very negative values may alter model evaluation. When working on a large set of catchments, they may also bias the calculation of the mean performance over the catchment set, by heavily weighting these outlier values.

15     Figure 3 shows that the catchments with negative KGE' values in Fig. 2 do not seem to exhibit any specific behaviour when evaluated with the KGE' values on untransformed flows: the criterion values are not lower in these catchments than in other





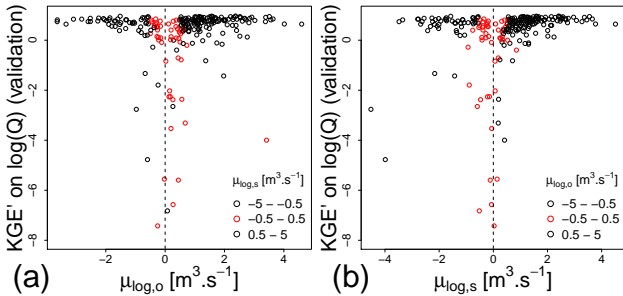

**Figure 2.** Values of KGE' on log-transformed flows versus the mean of the log-transformed observed (a) and simulated (b) flows. Each dot represents the performance obtained in validation for one catchment after calibration with the KGE' on untransformed flows as an objective function. The red dots represent the catchments where the average of the log-transformed simulated (a) or observed (b) flows is around 0.

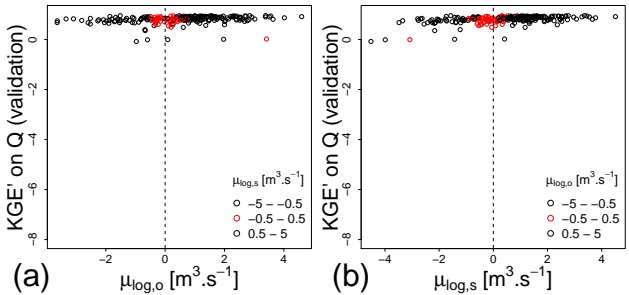

**Figure 3.** Values of KGE' on untransformed flows versus the mean of the log-transformed observed (a) and simulated (b) flows. Each dot represents the performance obtained in validation for one catchment after calibrating with the KGE' on untransformed flows as an objective function. The red dots represent the catchments where the average of the log-transformed simulated (a) or observed (b) flows is around 0.

catchments. Furthermore, this result can be completed by making the same plot for other transformations giving more weight on low flows. Figure 4 shows that square rooted (Fig. 4 (a) and (b)) and inverse (Fig. 4 (c) and (d)) transformations do encounter the same problems as with the logarithm for catchments that have an average log-transformed flow around zero.

The KGE' on log-transformed flows can also be compared to the NSE using the same transformation. Figure 5 shows that, when KGE' is significantly lower than NSE, the average of log-transformed flows (observed or simulated) is around zero (red dots in the figure). This tends to confirm that the strongly negative KGE' values stem more from a numerical issue than an actual problem in simulated values because the NSE values in these catchments remain correct.

In this technical note, the impact of a near-zero standard deviation of log-transformed flows is not presented because it is rarer than near-zero mean values. The standard deviations of flows on the catchments studied are indeed all significantly higher than zero.




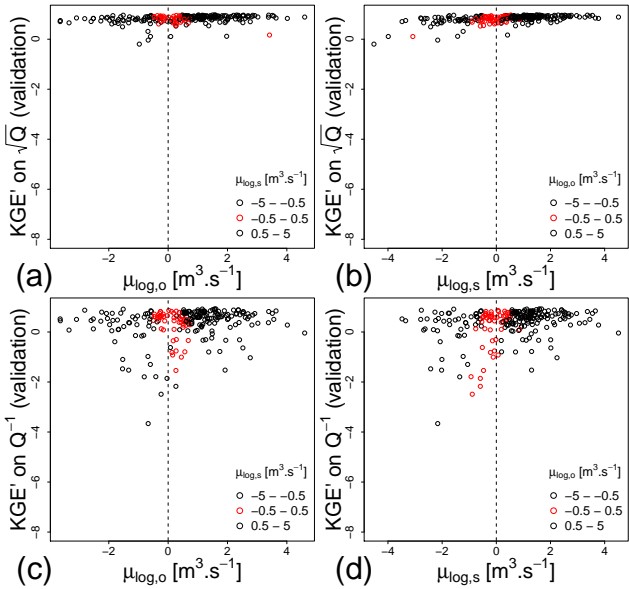

**Figure 4.** Values of KGE' on square root ((a) and (b)) and inverse ((c) and (d)) transformed flows versus the mean of the log-transformed observed and simulated flows. Each dot represents the performance obtained in validation for one catchment after calibration with the KGE' on untransformed flows as an objective function. The red dots represent the catchments where the average of the log-transformed simulated ((a) and (c)) or observed ((b) and (d)) flows is around 0.

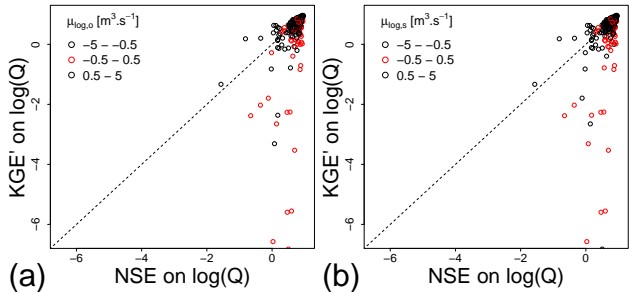

**Figure 5.** Comparison between KGE' and NSE values on the validation period using a calibration with KGE' on untransformed flows as an objective function. The red dots represent the catchments where the average of log-transformed observed (a) or simulated (b) flows is around 0.

## 5.2 Dependence on the flow unit chosen

The dependence of KGE' on log-transformed flows on the chosen flow units can easily be shown by plotting the KGE' on log-transformed flows in cubic metres per second versus the KGE' on log-transformed flows in litres per second. Figure 6 (b) shows that, for the catchments tested, the values of KGE' on log-transformed flows clearly depend on the flow unit used. A more optimistic evaluation of model performance will generally be obtained with the flows in $l \cdot s^{-1}$. As a comparison, Fig. 6





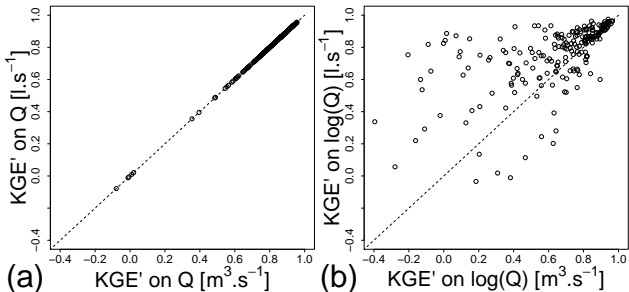

**Figure 6.** Dependence on flow units of the KGE' using untransformed flows (a) and log-transformed flows (b) on the 240 catchments. The parameters used for simulation evaluation were obtained by calibrating GR4J using KGE' on untransformed flows.

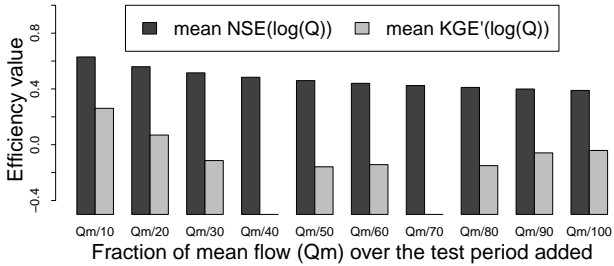

**Figure 7.** Sensitivity of NSE and KGE' to the fraction of average flows that is added to flows to avoid zero flows in the logarithmic transformation for 240 catchments over the validation period. This graph is inspired by Fig. 9 in Pushpalatha et al. (2012).

(a) shows that the KGE' with untransformed flows is not affected by the flow unit change. This dimension dependence makes the KGE' values based on log-transformed flows very difficult to interpret.

The more optimistic results using $l \cdot s^{-1}$ than using $m^3 \cdot s^{-1}$ can be explained analytically. Considering Eq. 5, a value of $\log(1000)$ is added to the $m^3 \cdot s^{-1}$ average. Because $\log(1000)$ is not negligible compared to the averages, adding it would

5     artificially improve $\beta$ and $\gamma$ and, by extension, the KGE' value.

### 5.3   Dependence on the value added to avoid the zero-flow issue

Pushpalatha et al. (2012) showed that the sensitivity of the NSE criterion on log-transformed flows to the small added constant declines when this constant decreases (from one-tenth to one-hundredth of the mean observed flow) and becomes limited for very small values  (see Fig. 9 in Pushpalatha et al., 2012). We performed the same test with the KGE' criterion and we obtained

10     a very different result (Fig. 7). The impact on performance is erratic for different values added to flows and does not show any trend. This may be due to the numerical issues shown in Sec. 5.1. For these reasons, the impact of added values can be major and may alter the model evaluation.





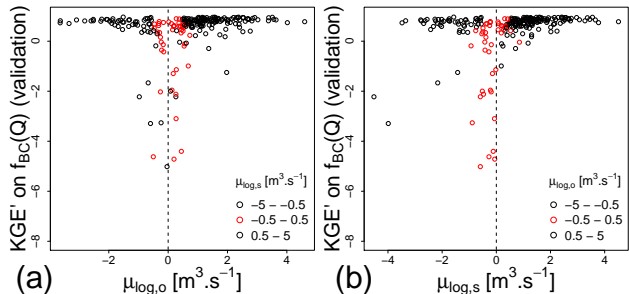

**Figure 8.** Values of KGE' on Box-Cox transformed flows versus the mean of the log-transformed observed (a) and simulated (b) flows. Each dot represents the performance obtained in validation for one catchment after calibration with the KGE' on untransformed flows as an objective function. The red dots represent the catchments where the average of the log-transformed simulated (a) or observed (b) flows is around 0.

## 5.4 The case of the Box-Cox transformation

As presented in Sect. 3.3, instead of adding a small value to flows, a Box-Cox transformation can be applied to flows to mimic the logarithm transformation without the zero-flow problem. However, even though it removes the dependence of the KGE' value to the value added to avoid zero flows, the other issues presented in the previous sections exist as for the logarithm.

For catchments in which the log-transformed flows' average is close to zero, the Box-Cox transformed flows exhibit the same behaviour as with the logarithm (Fig. 8). This result is logical because the Box-Cox transformation of 1 is equal to 0, as for the logarithmic transformation.

The Box-Cox transformation is also dependent on the units (Fig. 9 (a)). However, for this last issue, a slight modification of the Box-Cox formula allows one to address this problem. The classical Box-Cox transformation can be written as:

$$f_{BC}(Q) = \frac{Q^{\lambda} - 1}{\lambda} \tag{6}$$

in which $\lambda$ is an exponent to be chosen by the user, $Q$ is the flow value for any unit and $f_{BC}$ is the Box-Cox function.

Using this equation, the KGE' on transformed flows will be unit-dependent because of the additive term 1 in the numerator. To avoid this, we can slightly modify the formula, by replacing the term 1 by a constant with a unit dependence (here we propose the hundredth of the mean flow) and by putting it to the power $\lambda$:

$$f'_{BC}(Q) = \frac{Q^{\lambda} - (0.01\mu_o)^{\lambda}}{\lambda} \tag{7}$$

Using Eq. 7, the KGE' criterion remains dimensionless using the Box-Cox transformation (Fig. 9 (b)).

Furthermore, because the zero of the modified Box-Cox function is not 1 any more, this transformation would reduce the issue of strongly negative values when $\mu_{\log,o}$ or $\mu_{\log,s}$ are around zero. However, there still is an issue if the average of simulated flows is around the zero of the modified Box-Cox function (i.e. if $\mu_s = (0.01 * \mu_o)^{\lambda}$, Fig. 10). This instability occurs

more rarely than for the logarithm transformation but can be more frequent if bigger percentage of the average of observed flow or different $\lambda$ value are used. Because this instability is due to $\mu_s$, it will only affect the KGE' (not the KGE).





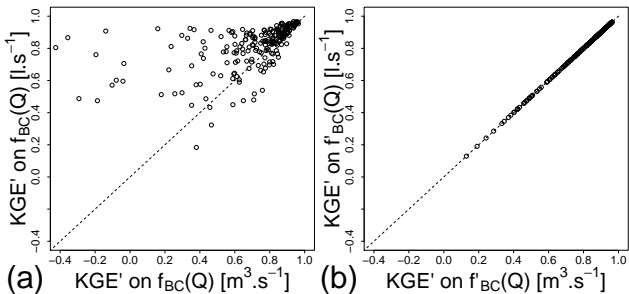

**Figure 9.** Dependence on flow units of the KGE' using Box-Cox transformed flows without adaptation ((a), Eq. 6) and with adaptation ((b), Eq. 7) on the 240 catchments. The parameters used for simulation evaluation were obtained by calibrating GR4J using KGE' on untransformed flows.

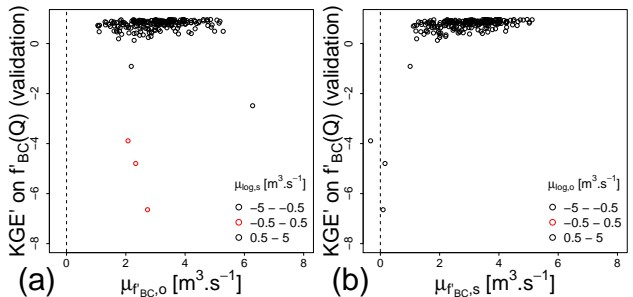

**Figure 10.** Values of KGE' on modified Box-Cox transformed flows (Eq. 7) versus the mean of this transformed observed (a) and simulated (b) flows. Each dot represents the performance obtained in validation for one catchment after calibration with the KGE' on untransformed flows as an objective function. The red dots represent the catchments where the average of the transformed simulated (a) or observed (b) flows is around 0.

The modified Box-Cox transformation (Eq. 7) allows to avoid unit dependence and to reduce the instability issues due to the values of average flows (especially when using the KGE). The behaviour of this modified transformation also remains similar to the one of the initial Box-Cox transformation except when $\mu_{\log,o}$ or $\mu_{\log,s}$ are around zero (Fig. 11).

# 6 Summary

## 6.1 Log transformation should not be used in the KGE or KGE' criterion

Given the previous results, we can argue that using log-transformed flows to calculate the KGE or the KGE' criterion can lead to difficulties in the interpretation of criterion values. The criterion does not remain dimensionless like NSE with a prior logarithmic transformation. It also becomes overly sensitive when the log-transformed flows' average becomes close to zero, yielding potentially very negative values, or when a small constant is added to flows prior to logarithmic transformation to cope with zero flows. Because of all these issues, logarithmic transformation should be avoided when using KGE'.





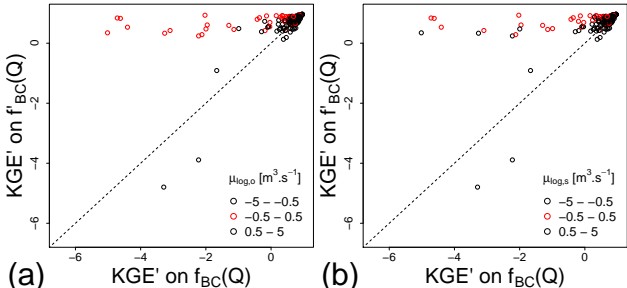

**Figure 11.** Comparison between KGE' values on Box-Cox and modified Box-Cox transformed flows on the validation period using a calibration with KGE' on untransformed flows as an objective function. The red dots represent the catchments where the average of log-transformed observed (a) or simulated (b) flows is around 0.

**Table 1.** Pros (+) and cons (-) of different flow transformations to improve consideration of low flows in KGE'

| Flow transformation | Decrease high-flow weight | Increase low-flow weight | No issue with zero flows | Dimensionless | No issue when flows average around 1 |
|---|---|---|---|---|---|
| Square root | + | - | + | + | + |
| Inverse | + | + | - | + | + |
| Logarithm | + | + | - | - | - |
| Box-Cox | + | + | + | + (if using Eq. 7) | + (if using Eq. 7) |

## 6.2 Alternatives

Instead of KGE' on log-transformed flows, several transformations can be used to calculate KGE'. The pros and cons for several transformations are summarised in Table 1. Regarding this table, the modified Box-Cox transformation (Eq. 7) seems to be the best solution but it still faces instabilities for some flow average values (for the KGE'). Thus, there is no ideal solution to avoid all problems. Modellers have to make a choice depending on their specific applications. Garcia et al. (2016), for example, recommend averaging two KGE' criteria computed on untransformed and inverted flows, into a composite criterion.

Note that many studies use NSE on log-transformed flows (see for example Lyon et al., 2017; Nguyen and Dietrich, 2018). Fortunately, the mathematical formulation of NSE avoids all the problematic aspects identified for KGE with the logarithmic transformation. However, this may not be a sufficient argument to continue to use NSE given the issues presented by Gupta et al. (2009) and Schaefli and Gupta (2007):

- the underestimation of variability,

- the low weight of water balance errors for catchments with highly variable flows,



– the poor benchmark represented by the mean flows for catchments with highly variable flows.

## 6.3 Final remarks

Two additional remarks should be taken into account on this topic. First, as noted by H. Kling in a personal communication,

prior transformations on flows in KGE (or in NSE) lead to a misinterpretation in the estimation of the water balance. The other components of the KGE also lose their initial physical meaning. KGE on transformed flows can give more information on low flows, but the physical interpretation of the criterion is not as simple as in the case of untransformed flows.

Secondly, even if it did not occur in our experiment, the issue described in this technical note may lead to problems during the calibration process. Indeed, it can create a strongly negative zone in the objective function hyperspace, which may negatively

impact the performance of local calibration algorithms.

*Author contributions.* L. Santos made the technical development and the analysis. The manuscript was written by him, G. Thirel and C. Perrin.

*Competing interests.* The authors declare that they have no conflicts of interest.

*Acknowledgements.* The authors thank Météo France for providing the data used in this work. We also wish to thank Alban De Lavenne,

Laure Lebecherel, Maria-Helena Ramos and Cedric Rebolho for the discussions on the different aspects of the issues using the logarithmic transformation with KGE. We thank Andrea Ficchí for his work on the database and Linda Northrup for her correction of the English language of the manuscript. Finally, we extend our thanks to Harald Kling for discussions on this issue.





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
