# Peer review of "Technical note: Pitfalls in using log-transformed flows within the KGE criterion"

_Hydrology and Earth System Sciences, 2018_

## Short Comment (SC1) · 9 Jun 2018

On Table 1: A fifth transformation of the low flow

A combination of the first two transformations, the square root and the inverse, leads back in time to the 1960s to a RoSR (reciprocal of the square root) transformation of the dry weather flow.

This physics-based double transformation was arrived at independently by Chapman (1964) in Australia, Ishihara and Tagaki (1965) in Japan, and this writer, Ding (1966) in Canada. All this, along with the fourth statistical one, the 1-parameter Box-Cox (1964) transformation, appeared in a 3-year time span from 1964 to 1966.

I'll be interested in the authors' view on this parametric-free RoSR transformation.

[Figure]

**References**

Chapman, T. G.: Effects of ground-water storage and flow on the water balance, in: Proceedings of "Water resources, use and management", Symposium held at Canberra by Australian Academy of Science, 9–13 September 1963, Melbourne Univ. Press, Victoria, 291–301, 1964.

Ding, J. Y.: Discussion of "Inflow hydrographs from large unconfined aquifers", by Ibrahim, H. A. and Brutsaert, W., J. Irrig. Drain. Div. Am. Soc. Civ. Eng., 92, 104–107, 1966.

Ishihara, T. and Takagi, F.: A study on the variation of low flow, Bulletin, Disaster Prevention Research Institute, Vol. 15, Part 2, No. 95, Kyoto Univ., Kyoto, Japan, 1965.

---

## Author Comment (AC1) · 14 Jun 2018

We would like to thank Dr J. Ding for reading our manuscript and for his comment.

We had not heard before about this transformation which is, actually, very interesting.

Regarding only Table 1, the inverted square root transformation shows exactly the same pros and con as the inverse transformation. It allows to decrease high-flow weight and increase low-flow weight in the KGE' calculation. A KGE' calculated on this transformation is also dimensionless and shows no issue when the flow average is around 1 (see Fig. 1 of this comment) and, as for inverse transformation, the inverted square root one needs specific attention for zero flows.

However, if we only consider numerical characteristics, the inverted square root transformation presents two advantages compared to invert transformation. The first one is that, even if it is sensitive to the constant added to avoid zero flows, this sensitivity is lower than the inverse transformation's sensitivity (as shown in Fig. 2). The second one is that the inverse transformation can be very extreme and totally erase the weight of high flows. The inverted square root can be seen as "smoother" than inverse.

In a nutshell, we consider the inverted square root transformation as a good compromise to replace logarithm transformation. We are grateful to Dr J. Ding for his suggestion and we propose to add this transformation in Table 1 and to add comments in Sect. 6.2. Obviously, we will acknowledge his contribution in the text.

Léonard Santos, on behalf of co-authors

PS: We only managed to find the Ishihara and Takagi reference but we are interested in reading the two other cited references. We would be grateful if you accept to send us a copy of these two works.
* * *
[Figure]

**Fig. 1.** Values of KGE' on inverse square rooted transformed flows versus the mean of the log-transformed observed (a) and simulated (b) flows. Each dot represents the performance obtained in validation.

[Figure]

[Figure]

**Fig. 2.** Sensitivity of KGE' to the fraction of average flows that is added to flows to avoid zero flows in the inverted square root transformation for 240 catchments over the validation period.

---

## Short Comment (SC2) · 17 Jun 2018

A graphical illustration

It's most gratifying to read the positive response from the authors to my suggestion of looking at a classical RoSR (i.e., ISR, the inverse square root) transformation.

Their discussion paper has appeared at a most opportune time, as I have ready a slide presentation on the Budyko evapotranspiration framework. This happens to include an illustration of the log- and RoSR-transformations of a dry weather flow hydrograph.

Two of these slides are shown here, Figure 9 and (Section) 7. In the latter, the storage exponent $N$ appears in a nonlinear storage-discharge relation, $Q \sim S^N$.

PS. I could supply my own 1966 ASCE discussion paper which should be available on an inter-library loan as I used to know. But I think it'll be worth their while to get and read a complete set of the referenced works. They can view my "Johnny came late" contribution in the context of the development of prevailing scientific thoughts. For example, why the Box-Cox (1964) transformation has gone mainstream in hydrology, but the ISR (1964-66) has not, as if some of us hadn't tried or hard enough. More food for thought.

[Figure]

[Figure]

Figure 9. Observed hydrograph and the transformed recession hydrographs for Crnojevica Spring, Bosnia and Herzegovina. (Source of flow data: Bonacci, 1993).

Crnovejica Spring, Bosnia and Herzegonia

$Q$

10 x $logQ$      R = -0.921

-100 x $(1/\sqrt{Q})$      R = 0.983

Discharge $Q$ (m$^3$ s$^{-1}$)

Time $t$ (d)

**Fig. 1.**

[Figure]

**7. The storage exponent $N$ - an example**

Figure 9 is an example for Crnovejica Spring, Bosnia and
Herzegovina. It shows the observed hydrograph (Bonacci, 1993).
In addition, it shows the log- and RoSR (Reciprocal of the Square
Root)-transformed recession hydrographs. The corresponding
storage exponent values are $N = 1$ and $N = 2$.

For clarity, the ordinates of the log-transformed hydrograph (shown
in green squares) are multiplied by a factor of 10, and those of
RoSR one (in red) by that of -100.

For log and RoSR transform, the **correlation coefficient**, slope
and intercept of the best-fit line are (**-0.921**, -0.228, 4.380) and
(**0.983**, 0.038, 0.009), respectively. The RoSR transform thus fits
the recession limb better than the log-transform.

[Figure]

**Fig. 2.**

---

## Short Comment (SC3) · 18 Jun 2018

Addendum

Unbeknown to me until now, Figure 9 hints at a new data transformation targeting hydrographs. This is called the NISR, the negative inverse square-root transformation of the flow $Q$: $-\frac{1}{\sqrt{Q}}$.

---

## Referee Comment (RC1) · L.A. Melsen (Referee) · 22 Jun 2018

Santos et al. present a highly relevant technical note about the application of log-transformations in combination with the Kling-Gupta Criterion. The manuscript is well written, well-structured, and contains a relevant example to demonstrate their point. Please find below my (mainly minor) comments.

"Major"

P. 6 l. 3-4 "Furthermore, this result can be completed by making the same plot

for other transformations giving more weight on low flows. Figure 4 shows that square rooted (Fig. 4 (a) and (b)) and inverse (Fig. 4 (c) and (d)) transformations do encounter the same problems as with the logarithm for catchments that have an average log-transformed flow around zero." This statement is inconsistent with the figures. The square root transformation does show a completely different pattern. Please clarify.

Related to that; table 1 states that square root transformation does not increase low flow weight, but to me it seems that it diminishes the weight of high flows, thereby somehow increasing the weight of low flows. Please clarify.

p. 8 l. 3 I understand "optimistic" refers to a higher model performance for KGE' when evaluated in l/s compared to m3/s. However, I don't really understand how Eq. 5 automatically implies this. In Eq. 5 I see that log(1000) is always added, but whether this leads to an improved or decreased model performance seems to me dependent on the bias in the model. Please clarify.

Minor

p. 4 l. 13 It would be good if the order of Box-Cox and adding a constant is changed in order to be consistent with results.

Figure 2 is relevant and insightful, but it takes some time to understand all information. Perhaps, it can be stressed in the caption that left, simulated in shown in red and right observed is shown in red (as is also done for a figure later in the manuscript).

p. 6 l. 7 "remain correct". Correct seems a vague term in this context (what is a correct objective function value?). Please consider rewording.

Consider to include the original KGE equation in Section 2 as well, especially because this information is relevant in the discussion of the modified Box-Cox. E.g. p. 9 l. 20, it will not affect the KGE because $\mu_s$ is not in the denominator in the original KGE (perhaps help the reader on this as well, e.g. on p. 10 just above the section Summary).

p. 3 l. 19 conversation -> conversion
* * *

---

## Author Comment (AC2) · 29 Jun 2018

We would like to thank Dr Lieke Melsen for her detailed analysis of the article. It helped us to clarify some points in the manuscript.

Major

- Page 6, line 3-4: "Furthermore, this result can be completed by making the same plot for other transformations giving more weight on low flows. Figure 4 shows that square rooted (Fig. 4 (a) and (b)) and inverse (Fig. 4 (c) and (d)) transformations do encounter the same problems as with the logarithm for catchments that have an average log-transformed flow around zero." This statement is inconsistent with the figures. The square root transformation does show a completely different pattern. Please clarify

This statement is clearly inconsistent as we forgot a word. Instead of "do encounter", we meant "do not encounter". We apologize for this mistake that totally change the meaning of the sentence.

However, we can discuss a little this point: Dr Lieke Melsen is right when saying that the square rooted flows show a completely different pattern but, to a lesser degree, it is also the case for the inverted flows. Indeed, if the KGE' on inverted values shows negative values for catchments that have an average log-transformed flow around zero, it also shows negative values for a significant part of the other catchments (Fig. 4 (c) and (d)). These negative values are more due to the difference between inverted flows and the untransformed flows than to some numerical flaws in the KGE'.

- Related to that; table 1 states that square root transformation does not increase low flow weight, but to me it seems that it diminishes the weight of high-flows, thereby somehow increasing the weight of low flows. Please clarify.

The reviewer is right, by decreasing the high-flows weight, the square root transformation indirectly increases the low-flows weight. We stated this for the square root transformation in order to highlight the fact that this transformation increases low-flow weights to a lesser extent compared to the inverse, Box-Cox or logarithmic transformations.

Instead of using two columns, namely about low and high flows columns, we propose to keep only one column named "Increases low-flow weight" and to use a different number of + signs as an intensity representation (+ for square root, ++ for logarithm, Box-Cox and the inverted square root, added following Dr John Ding comment and +++ for inverse).

- Page 8, line 3: I understand "optimistic" refers to a higher model performance for

KGE' when evaluated in l/s compared to m3/s. However, I don't really understand how Eq. 5 automatically implies this. In Eq. 5 I see that log(1000) is always added, but whether this leads to an improved or decreased model performance seems to me dependent on the bias in the model. Please clarify.

To clarify the impact of the added $\log(1000)$ we can calculate the bias ratio of log-transformed flows in $\mathrm{l} \cdot \mathrm{s}^{-1}$ regarding the average of log-transformed flows in $\mathrm{m}^3 \cdot \mathrm{s}^{-1}$. Using Eq. 5, the bias ratio is equal to:

$$\beta_{\log}[\mathrm{l} \cdot \mathrm{s}^{-1}] = \frac{\log(1000) + \mu_{\log,\mathrm{s}}[\mathrm{m}^3 \cdot \mathrm{s}^{-1}]}{\log(1000) + \mu_{\log,\mathrm{o}}[\mathrm{m}^3 \cdot \mathrm{s}^{-1}]} \tag{1}$$

In the tested data set, $\mu_{\log,\mathrm{s}}[\mathrm{m}^3 \cdot \mathrm{s}^{-1}]$ and $\mu_{\log,\mathrm{o}}[\mathrm{m}^3 \cdot \mathrm{s}^{-1}]$ (the log-transformed flows averages of respectively the simulated and observed flows in $\mathrm{m}^3 \cdot \mathrm{s}^{-1}$) are in majority between $-4$ and $4$. Because $\log(1000)$ is higher than the flows averages ($\approx 6.9$), it will have a greater impact on the ratio calculation than the average flow itself which leads to a tendency to improve the ratio. As we use the KGE', the $\gamma$ ratio is also affected and, because of the interaction between the average and the standard deviation of flows it is even more complicated to predict the ratio difference between $\mathrm{m}^3 \cdot \mathrm{s}^{-1}$ and $\mathrm{l} \cdot \mathrm{s}^{-1}$.

To illustrate this on the data set used in the article we plotted the values of the three KGE' components for the log-transformed flows in $\mathrm{m}^3 \cdot \mathrm{s}^{-1}$ and $\mathrm{l} \cdot \mathrm{s}^{-1}$ (Fig. 1 of this answer). Fig. 1 (c) shows that the bias ratio tends to be improved in $\mathrm{l} \cdot \mathrm{s}^{-1}$ especially for the catchments that have a bad bias ratio in $\mathrm{m}^3 \cdot \mathrm{s}^{-1}$. The difference between the two flow units is more complicated in the case of the coefficient of variation ratios (Fig. 1 (b)).

In a nutshell, the KGE' value tends to be higher because of the artificial improvement of the bias ratio but the coefficients of variation ratio can vary differently and lead to a decrease of the KGE' value.

In the manuscript, we will replace "optimistic" by "higher model performances" and add some sentences to better explain the reasons of this apparent improvement of performances.

Minor

- Page 4, line 13 It would be good if the order of Box-Cox and adding a constant is changed in order to be consistent with results.

  It will be done.

- Figure 2 is relevant and insightful, but it takes some time to understand all information. Perhaps, it can be stressed in the caption that left, simulated in shown in red and right observed is shown in red (as is also done for a figure later in the manuscript).

  Regarding this remark, we propose to replace " The red dots represent the catchments where the average of the log-transformed simulated (a) or observed (b) flows is around 0." by "In plot (a), the axis values represents the observed log-transformed flows averages and the color represents the simulated ones while in the plot (b), it is the opposite."

- Page 6, line 7 "remain correct". Correct seems a vague term in this context (what is a correct objective function value?). Please consider rewording.

  The reviewer is right, the word "correct" is not well chosen, particularly because some of the NSE values in question are around zero which denotes a bad simulation. It will be replaced by "positive or around zero".

- Consider to include the original KGE equation in Section 2 as well, especially because this information is relevant in the discussion of the modified Box-Cox. E.g. p. 9 l. 20, it will not affect the KGE because $\mu_s$ is not in the denominator

in the original KGE (perhaps help the reader on this as well, e.g. on p. 10 just above the section Summary).

It is a good suggestion, an equation will be added replacing the $\gamma$ term of Eq. 1 by an $\alpha$. The $E_{\mathsf{KG}}$ in Eq. 1 will be denoted $E'_{\mathsf{KG}}$ and proper reference to the KGE equation in page 9 and 10.

- Page 3, line 19 conversation -> conversion

It will be fixed

Léonard Santos, on behalf of co-authors

[Figure]

**Fig. 1.** Values difference between cubic metres and litres per seconds of the three components of the KGE' calculated on log-transformed flows.

---

## Referee Comment (RC2) · B. Guse (Referee) · 2 Jul 2018

Review of the technical note "Pitfalls in using log-transformed flows within the KGE criterion" by Santos et al.

In this technical note, Santos et al. shows that the application of KGE is limited for log-transformed data. They shows some options to handle with this limitation. Santos et al. presented very clearly why the KGE is limited for log-transformed data. I like this technical note and have only some minor comments.

Minor comments:

P. 5, L. 28: The modelling period is subdivided into a calibration (2003-2008) and a validation period (2008-2013). Please make clear whether the year 2008 belongs to

[Figure]

the calibration or validation period.

Figs. 2-11: If possible, I recommend to add a header to the subplots. Since all these figures have a similar layout, it would be good to differentiate them in a clearer way. E.g. you may add "observations" and "simulations" as header to the subplots of Fig. 3.

Fig. 2-3: Maybe you can merge both figures by adding the information that you compare log-transformed and untransformed data.

Technical comments:

I recommend to avoid paragraphs with only one or two sentences such as on P.4, L. 20-27.

P. 13, L. 4: Please remove "pp." in the Coron et al.

---

## Short Comment (SC4) · 8 Jul 2018

1) From the reader's point of view, considering the current discussion, the current version of the manuscript needs to be reviewed by qualified referees who are specialized in the subject that is presented in the manuscript. Since this is a technical note, the need for more technical evaluation by the referees is required. Moreover, the referees' role should not be harbored to state whether or not the referees "like/hate/love" the manuscript.

2) As per the authors [see P-2 LN-2], Gupta et al. (2009) clearly demonstrated that discharge variability is not correctly taken into account for the evaluation. Therefore, Gupta et al. (2009) proposed a new criterion, Kling-Gupta efficiency (KGE), which provides

direct assessment of four aspects of discharge time series, namely shape and timing, water balance and variability[see P-2 LN-6]. As far as I remember, in 2006, this piece of idea was introduced by a graduate student. Therefore, respecting Mr. Donald Trump's intention of preventing people from stealing someone's ideas/works/technologies, it would be more appropriate for the authors to evaluate the originality of Gupta et al. (2009)'s work.

3) As per the authors, The KGE' criterion (Kling et al., 2012, denoted EKG in Eq. 1) is written as a sum of the distances to 1 (perfect value) of three components of the modelling error [see P-2 LN-20]. What is meant by "sum" of the "distances" to 1? What is the mathematical formula that is used to compute the distance? If we consider a three dimensional space (i.e., x-axis=ratio-1, y-axis=ratio-2, z-axis=ratio-3), isn't the square root component merely the distance from the origin (i.e., [1, 1, 1])?

4) What is the physical meaning of equation (1)? Let's say that the right-hand side of the equation (1) has two components. The first component is "1". The second component is the square root component that includes the ratios (e.g., beta). Why would you subtract the second component (i.e., square root component) from the first component (i.e., 1)? What is the physical meaning of the second component? What is the physical meaning of the first component? If the second component of the equation (1) represents the distance (see the definition), as per dimensional theories, the first component needs to be a distance. Otherwise, the operator (i.e., negative sign) becomes meaningless. What is the distance represented by the first component? What is the origin for the distance that is represented by the first component?

5) Does your equation (3) evaluate the water balance error? What is meant by water balance? What is the range of your beta value? Assume that we have the following monthly observed flow values :5,5,4,5,5,5,5,5,6,5,5,5. Assume that we have the following monthly simulated flow values: :5,5,6,5,5,5,5,5,5,5,5,4.As per your equation(3), if we consider all the flow values, the beta value is 1.However, if we consider the first six months, the value of the beta is not equal to 1. Is your beta value time dependent?

[Figure]

6) Assume that the ratio-1=1(i.e., equation (2)), ratio-2=1(i.e., equation (3)), and ratio-3=0.5(i.e., equation (4)). As per your equation (1), the value of KGE is 0.5. Now, assume that the ratio-1=1(i.e., equation (2)), ratio-2=0.5(i.e., equation (3)), and ratio-3=1(i.e., equation (4)). As per your equation (1), the value of KGE is 0.5. What is the physical meaning of the KGE values?

7) As per your equation (4), the ratio-3 is a function of your beta value. In other words, your ration-3 is a function of ratio-2(i.e., equation (3)). This gives an indication that the ratio-3 that is accounted in your equation (1) repeats the influence of ratio-2 in equation (1).

---

## Editor Comment (EC1) · B. Schaefli (Editor) · 9 Jul 2018

Dear Colleague

I appreciate your active contribution to the public discussion of this manuscript. However, I think that in the future, you might want to adapt your comment writing style to such a public discussion.

You question the qualification of the reviewers that I have invited to review this paper ("manuscript needs to be reviewed by qualified referees"), which I can accept (even if I do not agree). **This does however not belong to a public discussion**. Furthermore, a technical note does not imply a different review process from any other paper, it is a different manuscript type ("this is a technical note, the need for more technical

evaluation by the referees is required")

And I would like to underline that **a constructive public review process** can indeed by accompanied by statements about whether a reviewer likes a paper. In any case, it is certainly not the role of a public comment to tell colleagues how they should review a paper.

Your short comment also implies that there are some open questions about the original idea of the KGE efficiency criterion. ("As far as I remember, in 2006, this piece of idea was introduced by a graduate student. Therefore, respecting Mr. Donald Trump's intention of preventing people from stealing someone's ideas/works/technologies, it would be more appropriate for the authors to evaluate the originality of Gupta et al. (2009)'s work."). Again, this public discussion is certainly not the place to discuss this kind of issues.

Regarding your more technical comments, I invite the authors of the paper to answer your comments / questions.

---

## Editor Comment (EC2) · B. Schaefli (Editor) · 10 Jul 2018

As the editor of this paper, I would like to thank J. Ding for this fruitful discussion.

---

## Author Comment (AC3) · 17 Jul 2018

**Answer to the review comments by Björn Guse**

We would like to thank Dr Björn Guse for his review of the article. We will try to improve our manuscript according to his comments.

[Figure]

Minor comments:

- Page 5, line 28: The modelling period is subdivided into a calibration (2003-2008) and a validation period (2008-2013). Please make clear whether the year 2008 belongs to the calibration or validation period.

  Actually we made a mistake in this description. To be exact, the calibration period covers the period from July 2005 to June 2009 and the validation one covers the period from July 2009 to July 2013. We apologize for this mistake and will modify the article accordingly.

- Figs. 2-11: If possible, I recommend to add a header to the subplots. Since all these figures have a similar layout, it would be good to differentiate them in a clearer way. E.g. you may add "observations" and "simulations" as header to the subplots of Fig. 3.

  This is a good suggestion, it will help making the figures more understandable and complements Dr Lieke Melsen's minor comment number 2.

- Fig. 2-3: Maybe you can merge both figures by adding the information that you compare log-transformed and untransformed data.

  This is also a good idea as Fig. 2 has more value when compared to Fig. 3. We will do it.

Technical comments:

- I recommend to avoid paragraphs with only one or two sentences such as on P.4, L.20-27.

  We will try to aggregate the smallest paragraphs for a better understanding of the manuscript.

- Page 13, L. 4: Please remove "pp." in the Coron et al.

  It will be done

Léonard Santos, on behalf of co-authors

---

## Short Comment (SC5) · 27 Jul 2018

**Re-ranking the transformations**

The NISR (negative inverse square-root) transformation can be generalized to a NIR (negative inverse root) one defined below:

$$J_N(Q) = -\frac{1}{\sqrt[N]{Q}} = -\frac{1}{Q^{1/N}}, \tag{1}$$

$$J_0(Q) = \log Q, \tag{2}$$

Some may dismiss the $J_2$ transformation as simply a sign change of the classical ISR

(inverse square-root) one. Indeed they are correct. But as Leonardo Da Vince (1452 - 1519) once said, "Simplicity is the ultimate sophistication."

For example, $J_2$ happens to be a subset of the 1-parameter Box-Cox (1964) transformation (Eq. 6 in Santos et al. paper) when parameter $\lambda = -1/2$,

$$f_{BC}^{\lambda=-1/2}(Q) = \frac{Q^\lambda - 1}{\lambda} = 2(1 + J_2), \tag{3}$$

The difference between these two ISR-type transformed values is:

$$f_{BC}^{\lambda=-1/2} - J_2 = 2 + J_2 = 2 - \frac{1}{\sqrt{Q}}, \tag{4}$$

This has a maximum value of 2.

Figure 1 shows the modifications of the four transformation methods being considered in the authors' Table 1. These are the original logarithmic, a fixed-parameter Box-Cox, the inverse negated, and the square root both inverted and negated, and labelled $J_0$, $f_{BC}^{\lambda=-1/2}$, $J_1$, and $J_2$, respectively. All four transformation curves share the same inverted U-shape, being an advantage for comparison purposes. These show their relative impact on the transformed flow values, most obviously on the lower ends.
* * *
[Figure]

**Fig. 1.** Comparison of Box-Cox and NIR (negative inverse root) transformation methods.

---

## Editor Comment (EC3) · B. Schaefli (Editor) · 22 Aug 2018

I would like to thank the authors and the reviewers for the fruitful discussion of this manuscript. The formal reviewers recommend only minor corrections. The discussion with J. Ding led to interesting new ideas that I invite the authors to include in their revised paper along their comments in the public discussion.

---

## Author Response (AR1)

**Final response**

In the following, the specific Author Comments (ACs) previously provided to the Short Comments (SCs) and Referee comments (RCs) are given again (questions are in blue, authors' responses are in black). In addition, it is now specified, if relevant, where the modifications were implemented in the revised manuscript (version with no apparent modifications).

Finally, the current Final response also includes the revised version of the manuscript with apparent additions in blue and removals in red and crossed.

**Answer to the review comments by Dr Lieke Melsen**

We would like to thank Dr Lieke Melsen for her detailed analysis of the article. It helped us to clarify some points in the manuscript.

**Major**

• Page 6, line 3-4: "Furthermore, this result can be completed by making the same plot for other transformations giving more weight on low flows. Figure 4 shows that square rooted (Fig. 4 (a) and (b)) and inverse (Fig. 4 (c) and (d)) transformations do encounter the same problems as with the logarithm for catchments that have an average log-transformed flow around zero." This statement is inconsistent with the figures. The square root transformation does show a completely different pattern. Please clarify

This statement is clearly inconsistent as we forgot a word. Instead of "do encounter", we meant "do not encounter". We apologize for this mistake that totally change the meaning of the sentence.

However, we can discuss a little this point: Dr Lieke Melsen is right when saying that the square rooted flows show a completely different pattern but, to a lesser degree, it is also the case for the inverted flows. Indeed, if the KGE' on inverted values shows negative values for catchments that have an average log-transformed flow around zero, it also shows negative values for a significant part of the other catchments (Fig. 4 (c) and (d)). These negative values are more due to the difference between inverted flows and the untransformed flows than to some numerical flaws in the KGE'.

Added/Modified: Sect. 5.1, p 6, lines 11-12 (of the revised manuscript) Figure 3 shows that square rooted (Fig. 3 (a) and (b)) and inverse (Fig. 3 (c) and (d)) transformations do **not** encounter the same problems as with the logarithm for catchments that have an average log-transformed flow around zero.

• Related to that; table 1 states that square root transformation does not increase low flow weight, but to me it seems that it diminishes the weight of high-flows, thereby somehow increasing the weight of low flows. Please clarify.

The reviewer is right, by decreasing the high-flow weight, the square root transformation indirectly increases the low-flow weight. We stated this for the square root transformation in order to highlight the fact that this transformation increases low-flow weights to a lesser extent compared to the inverse, Box-Cox or logarithmic transformations.

Instead of using two columns, namely about low and high flows columns, we propose to keep only one column named "Increases low-flow weight" and to use a different number of + signs as an intensity representation (+ for square root, ++ for logarithm, Box-Cox and the inverted square root, added following Dr John Ding comment and +++ for inverse).

Added/Modified: We removed the column "Decrease high-flow weight" and used different numbers of + in the column "Increase low-flow weight" in table 1 (Sect. 6.2, top of p 12).

• Page 8, line 3: I understand "optimistic" refers to a higher model performance for KGE' when evaluated in 1/s compared to m3/s. However, I don't really understand how Eq. 5 automatically implies this. In Eq. 5 I see that log(1000) is always added, but whether this leads to an improved or decreased model performance seems to me dependent on the bias in the model. Please clarify.

To clarify the impact of the added  $\log(1000)$  we can calculate the bias ratio of log-transformed flows in  $1 \cdot s^{-1}$  regarding the average of log-transformed flows in  $m^3 \cdot s^{-1}$ . Using Eq. 5, the bias ratio is equal to:

$$\beta_{\log}[\mathbf{l} \cdot \mathbf{s}^{-1}] = \frac{\log(1000) + \mu_{\log,s}[\mathbf{m}^3 \cdot \mathbf{s}^{-1}]}{\log(1000) + \mu_{\log,o}[\mathbf{m}^3 \cdot \mathbf{s}^{-1}]}$$
(1)

In the tested data set,  $\mu_{\log,s}[m^3 \cdot s^{-1}]$  and  $\mu_{\log,o}[m^3 \cdot s^{-1}]$  (the log-transformed flows averages of respectively the simulated and observed flows in  $m^3 \cdot s^{-1}$ ) are in majority between -4 and 4. Because  $\log(1000)$  is higher than the flows averages ( $\approx 6.9$ ), it will have a greater impact on the ratio calculation than the average flow itself which leads to a tendency to improve the ratio. As we use the KGE', the  $\gamma$  ratio is also affected and, because of the interaction between the average and the standard deviation of flows it is even more complicated to predict the ratio difference between  $m^3 \cdot s^{-1}$  and  $l \cdot s^{-1}$ .

To illustrate this on the data set used in the article we plotted the values of the three KGE' components for the log-transformed flows in  $m^3 \cdot s^{-1}$  and  $l \cdot s^{-1}$  (Fig. 1 of this answer). Fig. 1 (c) shows that the bias ratio tends to be improved in  $l \cdot s^{-1}$  especially for the catchments that have a bad bias ratio in  $m^3 \cdot s^{-1}$ . The difference between the two flow units is more complicated in the case of the coefficient of variation ratios (Fig. 1 (b)).

In a nutshell, the KGE' value tends to be higher because of the artificial improvement of the bias ratio but the coefficients of variation ratio can vary differently and lead to a decrease of the KGE' value.

In the manuscript, we will replace "optimistic" by "higher model performance" and add some sentences to better explain the reasons of this apparent improvement of performances.

Added/Modified: Sect. 5.2, p 8, lines 8-13 The higher model performance when using  $l \cdot s^{-1}$  than when using  $m^3 \cdot s^{-1}$  can be explained analytically. Considering Eq. 7, the formula of the bias ratio in  $l \cdot s^{-1}$  regarding the averages in  $m^3 \cdot s^{-1}$  is:

$$\beta_{\log}[l \cdot s^{-1}] = \frac{\log(1000) + \mu_{\log,s}[m^3 \cdot s^{-1}]}{\log(1000) + \mu_{\log,o}[m^3 \cdot s^{-1}]}$$

Because  $\log(1000)$  is not negligible compared to the averages, adding this constant term would artificially improve  $\beta$  and, by extension, the KGE' value. The  $\gamma$  ratio is also affected and, due to the interactions between the standard deviation and the averages, modify differently the KGE' value.

**Minor**

• Page 4, line 13 It would be good if the order of Box-Cox and adding a constant is changed in order to be consistent with results.

It will be done.

Added/Modified: Done (see Sect. 3.3, p 4, lines 13-19)

• Figure 2 is relevant and insightful, but it takes some time to understand all information. Perhaps, it can be stressed in the caption that left, simulated in shown in red and right observed is shown in red (as is also done for a figure later in the manuscript).

Regarding this remark, we propose to replace "The red dots represent the catchments where the average of the log-transformed simulated (a) or observed (b) flows is around 0." by "In plot (a), the axis values represents the observed logtransformed flows averages and the color represents the simulated ones while in the plot (b), it is the opposite."

Added/Modified: We did this modification not only in Fig. 2 caption but also in the captions of the three other figures that are similar to Fig. 2 (namely Fig. 3, 7 and 9).

• Page 6, line 7 "remain correct". Correct seems a vague term in this context (what is a correct objective function value?). Please consider rewording.

The reviewer is right, the word "correct" is not well chosen, particularly because some of the NSE values in question are around zero which denotes a bad simulation. It will be replaced by "positive or around zero".

Added/Modified: Sect. 5.1, p 7, lines 1-2 This tends to confirm that the strongly negative KGE' values stem more from a numerical issue than an actual problem in simulated values because the NSE values in these catchments remain **positive** or around zero.

• Consider to include the original KGE equation in Section 2 as well, especially because this information is relevant in the discussion of the modified Box-Cox. E.g. p. 9 l. 20, it will not affect the KGE because  $\mu_s$  is not in the denominator in the original KGE (perhaps help the reader on this as well, e.g. on p. 10 just above the section Summary).

It is a good suggestion, an equation will be added replacing the  $\gamma$  term of Eq. 1 by an  $\alpha$ . The  $E_{\rm KG}$  in Eq. 1 will be denoted  $E'_{\rm KG}$  and proper reference to the KGE equation in page 9 and 10.

Added/Modified: Sect. 2, p 2, line 20 The KGE and KGE' criteria (Gupta et al., 2009; Kling et al., 2012, respectively denoted  $E_{\rm KG}$  and  $E'_{\rm KG}$  in Eq. 1 and Eq. 2)

Sect. 2, p 2, lines 23-24

$$E_{\rm KG} = 1 - \sqrt{(r-1)^2 + (\beta - 1)^2 + (\alpha - 1)^2}$$

$$E'_{\rm KG} = 1 - \sqrt{(r-1)^2 + (\beta-1)^2 + (\gamma-1)^2}$$

Sect. 2, p 3, lines 3-4 –  $\alpha$ , the ratio between the simulated and observed standard deviations evaluates the flows variability error:

$$\alpha = \frac{\sigma_{\rm s}}{\sigma_{\rm o}}$$

Sect. 5.4, p 10, lines 10-11 Because this instability is due to  $\mu_s$  (which is only in the denominator of the  $\gamma$  ratio in Eq. 6), it will only affect the KGE'. The KGE is not affected because an  $\alpha$  ratio is used instead of the  $\gamma$  ratio (Eq. 1 and 5)

• Page 3, line 19 conversation -> conversion

It will be fixed

Added/Modified: Sect. 3.2, p 3, lines 23-24 It can be easily demonstrated that  $\gamma$ ,  $\beta$  and r remain identical when flow is expressed in any of these two units, since the division by 1000 necessary for the **conversion** is eliminated in the ratios.

Figure 1: Values difference between cubic metres and litres per seconds of the three components of the KGE' calculated on log-transformed flows.

**Answer to the review comments by Dr Björn Guse**

We would like to thank Dr Björn Guse for his review of the article.

**Minor comments:**

• Page 5, line 28: The modelling period is subdivided into a calibration (2003-2008) and a validation period (2008-2013). Please make clear whether the year 2008 belongs to the calibration or validation period.

Actually we made a mistake in this description. To be exact, the calibration period covers the period from July 2005 to June 2009 and the validation one covers the period from July 2009 to July 2013. We apologize for this mistake and will modify the article accordingly.

Added/Modified: Sect. 4.1, p 5, lines 1-2 The availability of data covers the 2005-2013 period.

Sect. 4.2, p 5, lines 6-8 The available records are split into a calibration (from July 2005 to June 2009) and a validation (from July 2009 to July 2013) period following a standard split-sample test procedure (Klemeš, 1986).

• Figs. 2-11: If possible, I recommend to add a header to the subplots. Since all these figures have a similar layout, it would be good to differentiate them in a clearer way. E.g. you may add "observations" and "simulations" as header to the subplots of Fig. 3.

This is a good suggestion, it will help making the figures more understandable and complements Dr Lieke Melsen's minor comment number 2.

Added/Modified: We added these headers in Fig. 2, 3, 7 and 9.

• Fig. 2-3: Maybe you can merge both figures by adding the information that you compare log-transformed and untransformed data.

This is also a good idea as Fig. 2 has more value when compared to Fig. 3. We will do it.

**Added/Modified:** Done, we removed the former Fig.3 and added it to Fig.2 (c) and (d).

**Technical comments:**

• I recommend to avoid paragraphs with only one or two sentences such as on P.4, L.20-27.

We will try to aggregate the smallest paragraphs for a better understanding of the manuscript.

**Added**/**Modified:** Following this advice, we aggregated 4 couples of paragraphs. These aggregations are pictured by small red lines in the following marked-up manuscript version.  Page 13, L. 4: Please remove "pp." in the Coron et al. It will be done
 Added/Modified: Done

**Answer to the comments by S. Mylevaganam**

We would like to thank S. Mylevaganam for his comments on the technical aspects of our manuscript and the Associate Editor for framing our answer regarding the issues which are out of the scope of the technical discussion.

• 1) From the reader's point of view, considering the current discussion, the current version of the manuscript needs to be reviewed by qualified referees who are specialized in the subject that is presented in the manuscript. Since this is a technical note, the need for more technical evaluation by the referees is required. Moreover, the referees' role should not be harbored to state whether or not the referees "like/hate/love" the manuscript.

This comment is not about the paper content and we therefore refer to the Editor's reply.

• 2) As per the authors [see P-2 LN-2], Gupta et al. (2009) clearly demonstrated that discharge variability is not correctly taken into account for the evaluation. Therefore, Gupta et al. (2009) proposed a new criterion, Kling-Gupta efficiency (KGE), which provides direct assessment of four aspects of discharge time series, namely shape and timing, water balance and variability[see P-2 LN-6]. As far as I remember, in 2006, this piece of idea was introduced by a graduate student. Therefore, respecting Mr. Donald Trump's intention of preventing people from stealing someone's ideas/works/technologies, it would be more appropriate for the authors to evaluate the originality of Gupta et al. (2009)'s work.

The objective of the paper was to discuss a specific application of a criterion previously published in an international journal after peer-review (Journal of Hydrology) and which has been widely used then. If there is a possible concern about the KGE criterion, the reviewer should directly contact the authors or the editors of Journal of Hydrology. We agree with the Editor that this discussion and our article are not the right place to discuss this issue. We are not aware of the graduate student work mentioned by the reviewer, for which no detail is given and therefore we are not able to evaluate the originality of the work.

• 3) As per the authors, The KGE' criterion (Kling et al., 2012, denoted EKG in Eq. 1) is written as a sum of the distances to 1 (perfect value) of three components of the modelling error [see P-2 LN-20]. What is meant by "sum" of the "distances" to 1? What is the mathematical formula that is used to compute the distance? If we consider a three dimensional space (i.e., x-axis=ratio-1, y-axis=ratio-2, z-axis=ratio-3), isn't the square root component merely the distance from the origin (i.e., [1, 1, 1])?

These questions show that our sentence is not precise enough. Mathematically speaking, the KGE' is a linear transformation of the Euclidian distance from the ideal point (i.e., [1, 1, 1]) in the three-dimensional space defined by the three ratios (Eq. (2) to (4)). In Eq. (1), this Euclidian distance is represented by the square root component and the computed linear transformation of this distance is  $f: x \mapsto 1-x$ . This function is used to allow the KGE' to have the same range

of values as for the NSE. We thank the reviewer for pointing out this and will attempt to be more precise in the manuscript by modifying the page 2 line 20 sentence.

Added/Modified: Sect. 2, p 2, lines 20-22 The KGE and KGE' criteria (Gupta et al., 2009; Kling et al., 2012, respectively denoted  $E_{\text{KG}}$  and  $E'_{\text{KG}}$  in Eq. 1 and Eq. 2) are written as a linear transformation  $(f : x \mapsto 1 - x)$  of the Euclidian distance to an ideal value (i.e. [1,1,1]) in a three dimensional space defined by three components of the modelling error:

• 4) What is the physical meaning of equation (1)? Let's say that the right-hand side of the equation (1) has two components. The first component is "1". The second component is the square root component that includes the ratios (e.g., beta). Why would you subtract the second component (i.e., square root component) from the first component (i.e., 1)? What is the physical meaning of the second component? What is the physical meaning of the first component of the equation (1) represents the distance (see the definition), as per dimensional theories, the first component needs to be a distance. Otherwise, the operator (i.e., negative sign) becomes meaningless. What is the distance represented by the first component? What is the origin for the distance that is represented by the first component?

First of all, regarding the dimensional theories, the KGE' expression is right. Indeed, a Euclidian distance in a space of dimensions without units (it is the case of the three ratios that form the KGE') is dimensionless. Thus, linearly transforming this dimensionless Euclidian distance is not wrong mathematically speaking.

However, the choice of the transformation f can be discussed. As said in the answer of the reviewer comment 3), the distance is subtracted to 1 to have the same range of values as the NSE criterion. This is clearly due to legacy because a lot of rainfall-runoff modellers are used to the NSE and to analyse its values. This subtraction can be discussed as the Euclidian distance stands for itself as an evaluation criterion but, because the transformation of this distance in the KGE' is linear, the interpretation of the KGE' values remains the same as for the Euclidian distance. Consequently, it has no impact on the evaluation of the performance of the model.

Regarding the physical meaning of the KGE', we will answer in our response of comment 6).

- 5) Does your equation (3) evaluate the water balance error? What is meant by water balance? What is the range of your beta value? Assume that we have the following monthly observed flow values :5,5,4,5,5,5,5,5,6,5,5,5. Assume that we have the following monthly simulated flow values :5,5,6,5,5,5,5,5,5,5,5,4.As per your equation(3), if we consider all the flow values, the beta value is 1.However, if we consider the first six months, the value of the beta is not equal to 1. Is your beta value time dependent?

The beta ratio represents the quantitative aspect of the simulation. If it is greater than 1, the model overestimates the discharge and if it is lower than 1, the model underestimate the discharge. We agree that the value of beta depends on the time as it is the case of all the other ratios. We used a split-sample test to limit the impact of this time dependency.

To avoid misunderstanding we will replace "water balance" by "bias".

Added/Modified: Sect.2, p 3, line  $1 - \beta$ , the bias term, evaluates the bias between observed and simulated flows:

6) Assume that the ratio-1=1 (i.e., equation (2)), ratio-2=1 (i.e., equation (3)), and ratio-3=0.5 (i.e., equation (4)). As per your equation (1), the value of KGE is 0.5. Now, assume that the ratio-1=1 (i.e., equation (2)), ratio-2=0.5 (i.e., equation (3)), and ratio-3=1 (i.e., equation (4)). As per your equation (1), the value of KGE is 0.5. What is the physical meaning of the KGE values?

The physical meaning of the KGE value itself is not well defined. It is simply an aggregated representation of the model error over the studied period. To understand its value, the modeller needs to have a look on the three components of the criterion separately. This is stated in the Gupta et al. (2009) publication and it is often done by the KGE users (for example in the work of Ficchí et al., 2016, cited in the present manuscript). Moreover, depending of the modeller's objectives, Gupta et al. (2009) also proposed to weight each component of the KGE.

• 7) As per your equation (4), the ratio-3 is a function of your beta value. In other words, your ratio-3 is a function of ratio-2 (i.e., equation (3)). This gives an indication that the ratio-3 that is accounted in your equation (1) repeats the influence of ratio-2 in equation (1).

In the publication that introduces the KGE', Kling et al. (2012) stated that: "For the variability ratio  $\gamma$  we used  $\frac{CV_s}{CV_o}$  instead of  $\frac{\sigma_s}{\sigma_o}$ , which was proposed in the original version of the KGE-statistic (Gupta et al., 2009). This ensures that the bias and variability ratios are not cross-correlated, which otherwise may occur when e.g. the precipitation inputs are biased.". In other words if the bias ratio is bad, the ratio of standard deviation will also be affected. To avoid the impact of average discharge error on the variability component, the standard deviation ratio is normalised by the bias ratio.

Added/Modified: Sect.2, p 3, line 6 These coefficients of variation are used to avoid the impact of bias on the variability indicator (Kling et al., 2012):

**Answer to the comments by Dr John Ding**

**On Table 1: A fifth transformation of the low flow (comment SC1)**

A combination of the first two transformations, the square root and the inverse, leads back in time to the 1960s to a RoSR (reciprocal of the square root) transformation of the dry weather flow. This physics-based double transformation was arrived at independently by Chapman (1964) in Australia, Ishihara and Tagaki (1965) in Japan, and this writer, Ding (1966) in Canada. All this, along with the fourth statistical one, the 1-parameter Box-Cox (1964) transformation, appeared in a 3-year time span from 1964 to 1966. I'll be interested in the authors' view on this parametric-free RoSR transformation.

**A graphical illustration (comment SC2)**

It's most gratifying to read the positive response from the authors to my suggestion of looking at a classical RoSR (i.e., ISR, the inverse square root) transformation. Their discussion paper has appeared at a most opportune time, as I have ready a slide presentation on the Budyko evapotranspiration framework. This happens to include an illustration of the log- and RoSR-transformations of a dry weather flow hydrograph. Two of these slides are shown here, Figure 9 and (Section) 7. In the latter, the storage exponent N appears in a nonlinear storage-discharge relation,  $Q \approx S^N$ .

**Addendum (comment SC3)**

Unbeknown to me until now, Figure 9 hints at a new data transformation targeting hydrographs. This is called the NISR, the negative inverse square-root transformation of the flow  $Q: -\frac{1}{\sqrt{Q}}$ .

**Re-ranking the transformations (comment SC5)**

The NISR (negative inverse square-root) transformation can be generalized to a NIR (negative inverse root) one defined below:

$$J_N(Q) = -\frac{1}{\sqrt[N]{Q}} = -Q^{\frac{-1}{N}}$$
(2)

$$J_0(Q) = \log Q \tag{3}$$

Some may dismiss the  $J_2$  transformation as simply a sign change of the classical ISR (inverse square-root) one. Indeed they are correct. But as Leonardo Da Vince (1452-1519) once said, "Simplicity is the ultimate sophistication."

For example,  $J_2$  happens to be a subset of the 1-parameter Box-Cox (1964) transformation (Eq. 6 in Santos et al. paper) when parameter  $\lambda = -\frac{1}{2}$

$$f_{BC}^{\lambda = -\frac{1}{2}}(Q) = \frac{Q^{\lambda} - 1}{\lambda} = 2(1 + J_2)$$
(4)

The difference between these two ISR-type transformed values is:

$$f_{BC}^{\lambda = -\frac{1}{2}} - J_2 = 2 + J_2 = 2 - \frac{1}{\sqrt{Q}}$$
(5)

This has a maximum value of 2.

Figure 1 shows the modifications of the four transformation methods being considered in the authors' Table 1. These are the original logarithmic, a fixed-parameter Box-Cox, the inverse negated, and the square root both inverted and negated, and labelled  $J_0$ ,  $f_{BC}^{\lambda=-\frac{1}{2}}$ ,  $J_1$  and  $J_2$ , respectively. All four transformation curves share the same inverted U-shape, being an advantage for comparison purposes. These show their relative impact on the transformed flow values, most obviously on the lower ends.

**Answer to Dr J. Ding comment SC1**

We would like to thank Dr J. Ding for reading our manuscript and for his comment.

We had not heard before about this transformation which is, actually, very interesting.

Regarding only Table 1, the inverted square root transformation shows exactly the same pros and con as the inverse transformation. It allows to decrease high-flow weight and increase low-flow weight in the KGE' calculation. A KGE' calculated on this transformation is also dimensionless and shows no issue when the flow average is around 1 (see Fig. 2 of this comment) and, as for inverse transformation, the inverted square root one needs specific attention for zero flows.

However, if we only consider numerical characteristics, the inverted square root transformation presents two advantages compared to invert transformation. The first one is that, even if it is sensitive to the constant added to avoid zero flows, this sensitivity is lower than the inverse transformation's sensitivity (as shown in Fig. 3). The second one is that the inverse transformation can be very extreme and totally erase the weight of high flows. The inverted square root can be seen as "smoother" than inverse.

In a nutshell, we consider the inverted square root transformation as a good compromise to replace logarithm transformation. We are grateful to Dr J. Ding for his suggestion and we propose to add this transformation in Table 1 and to add comments in Sect. 6.2. Obviously, we will acknowledge his contribution in the text.

**Answer to Dr J. Ding comments SC2, SC3 and SC5**

First of all, we would like to thank again Dr J. Ding for his valuable contribution to the discussion.

• Regarding John Ding's SC3 comment and the use of negative or positive inverted root we can argue that, in the context of this manuscript, the change of sign has no impact. Indeed, following the example of the inverted flows  $(-J_1 \text{ in comment SC5})$ , the mean and the standard deviation are linked to the ones of  $J_1$  by the following relations, respectively:

$$\mu_{-Q^{-1}} = -\mu_{Q^{-1}}
\sigma_{-Q^{-1}} = \sigma_{Q^{-1}}$$
(6)

Figure 2: Values of KGE' on inverse square rooted transformed flows versus the mean of the log-transformed observed (a) and simulated (b) flows. Each dot represents the performance obtained in validation for one catchment after calibration with the KGE' on untransformed flows as an objective function. The red dots represent the catchments where the average of the log-transformed simulated (a) or observed (b) flows is around 0.

Figure 3: Sensitivity of KGE' to the fraction of average flows that is added to flows to avoid zero flows in the inverted square root transformation for 240 catchments over the validation period.

with  $\mu_{Q^{-1}}$  and  $\sigma_{Q^{-1}}$  respectively the mean and the standard deviation of inverted flows and  $\mu_{-Q^{-1}}$  and  $\sigma_{-Q^{-1}}$  respectively the mean and the standard deviation of negative inverted flows.

The consequence of Eq. 6 in this comment, using Eq. 2 to 4 in the manuscript is that:

$$\begin{aligned} r_{-Q^{-1}} &= r_{Q^{-1}} \\ \beta_{-Q^{-1}} &= \beta_{Q^{-1}} \\ \gamma_{-Q^{-1}} &= \gamma_{Q^{-1}} \end{aligned}$$
 (7)

and, using Eq.1 of the manuscript:

$$KGE'(-Q^{-1}) = KGE'(Q^{-1})$$
 (8)

The sign of the transformation has, thus, no importance in the KGE' (and also KGE) calculation. For this reason we will keep positive transformations in our manuscript.

We show the equivalence of KGE' values for both aforementioned transformations on our data set in Fig. 4 of this answer to Dr Ding's comment.

- The choice of N in the transformations  $J_N$  proposed by Dr J. Ding can be interesting. One may choose the N value according to the weight intended on low flows. The higher N, the lower the weight on low flows. In addition, regarding Dr. J. Ding comment SC2, the value of N can also be deduced from the observation of recession curve in the simulated catchment.
- Regarding comment SC5, we will add this generic inverted root transformation in table 1 of the manuscript instead of the inverted square root as we stated it in comment AC1. We will also add a comment on its parametrization in the text.
- About the correspondence between the inverted root transformation and the Box-Cox transformation, Dr J. Ding is right arguing that, if  $\lambda = -\frac{1}{N}$ , a linear relation links the Box-Cox transformation and the inverted root transformation. However, to obtain this linear relation,  $\lambda$  has to be negative and, as much as we know, hydrologists who use the Box-Cox transformation always use a positive  $\lambda$  value because it allows to avoid issues with zero flows. As a consequence, we will keep the Box-Cox transformation in the table as it is.

PS: As an answer of SC2 comment sentence "For example, why the Box-Cox (1964) transformation has gone mainstream in hydrology, but the ISR (1964-66) has not, as if some of us hadn't tried or hard enough.", we can hypothesize that the greater interest for Box-Cox is due to its property to avoid the zero-flow issue. However, it is also possible that the use of Box-Cox is also due to legacy as it is the case for NSE criterion.

Figure 4: Equality of the KGE' values of the GR4J simulations using inverse transformation and negative inverse transformation on the 240 tested catchments.

**Added/Modified**

Sect. 6.2, p 11, lines 3-7 The inverted root is an example of used transformation that is not tested in the article but leads to increase the weight of low flows (Chapman,

1964; Ishihara and Takagi, 1965; Ding 1966). It can be parametrised with the value of the power in the root  $(Q^{-\frac{1}{N}})$ . Depending on the value of N, there will be more or less weight on low flows. The higher N is and the less the weight on low flows is. This N value can also be determined with the recession curves of observed flows.

We add a line in Table 1 (top of p 12) to describe the pros and cons of the parametric inverted root proposed by Dr Ding.

**Technical note: Pitfalls in using log-transformed flows within the KGE criterion**

Léonard Santos, Guillaume Thirel, and Charles Perrin Irstea, HYCAR Research Unit, 1 rue Pierre-Gilles de Gennes, 92160 Antony, France *Correspondence to:* Léonard Santos (leonard.santos@irstea.fr)

Abstract. Log-transformed discharge is often used to calculate performance criteria to better focus on low flows. This prior transformation limits the heteroscedasticity of model residuals and was largely applied in criteria based on squared residuals, like Nash-Sutcliffe efficiency (NSE). In the recent years, NSE has been shown to have mathematical limitations and Kling-Gupta efficiency (KGE) was proposed as an alternative to provide more balance between the expected qualities of a model

5 (namely representing the water balance, flow variability and correlation). As in the case of NSE, several authors used the KGE criterion (or its improved version KGE') with a prior logarithmic transformation on flows. However, we show that the use of this transformation is not adapted to the case of the KGE (or KGE') criterion and may lead to several numerical issues, potentially resulting in a biased evaluation of model performance. We present the theoretical underpinning aspects of these issues and concrete modelling examples, showing that KGE' computed on log-transformed flows should be avoided. Alternatives are

10 discussed.

15

**1 Introduction**

In the context of rainfall-runoff modelling, evaluating the quality of the models' outputs is essential. Deterministic simulations are commonly evaluated using efficiency criteria such as Nash-Sutcliffe efficiency (NSE, Nash and Sutcliffe, 1970). The choice of the criteria obviously depends on the modeller's objective. For example, one may wish to focus on the overall water balance evaluation, or more specifically on the simulation of different flow ranges, typically high, intermediate or low flows. For these different objectives, given that the model residuals are generally not homoscedastic and often depend on the flow magnitude, one common option to focus more closely on specific flow ranges is to apply various prior transformations on the simulated

and observed discharge time series to distort the range of errors, which consequently changes the relative weight of different flow ranges in the criterion. This is commonly done within the NSE criterion, which has been one of the most popular criteria

20 used in hydrological modelling in the past few decades. NSE is the distance to 1 of the ratio between the mean square error of the model and the variance of observed flows. Compared to the basic criterion computed on untransformed flows, a prior squared transformation on flows would put even more weight on high flows, a logarithmic or inverse transformation would put more weight on low flows while a square root transformation would have an intermediate effect (Krause et al., 2005; Oudin et al., 2006; De Vos and Rientjes, 2010; Pushpalat